# Ultrastructure and Physiological Characterization of *Morchella* Mitospores and Their Relevance in the Understanding of the Morel Life Cycle

**DOI:** 10.3390/microorganisms11020345

**Published:** 2023-01-30

**Authors:** Wei Liu, Peixin He, Jin Zhang, Liyuan Wu, Lingfang Er, Xiaofei Shi, Zhijia Gu, Fuqiang Yu, Jesús Pérez-Moreno

**Affiliations:** 1Germplasm Bank of Wild Species, Yunnan Key Laboratory for Fungal Diversity and Green Development, Kunming Institute of Botany, Chinese Academy of Sciences, Kunming 650201, China; 2College of Food and Biological Engineering, Zhengzhou University of Light Industry, Zhengzhou 450002, China; 3College of Resource and Environment, Yunnan Agricultural University, Kunming 650100, China; 4Edafología, Campus Montecillo, Colegio de Postgraduados, Texcoco 56230, Mexico

**Keywords:** morel cultivation, edible mushrooms, mating types, conidia, morel life cycle, ultrastructure, gamete, spermatia

## Abstract

Morels, which belong to the Ascomycete genus *Morchella*, are highly valued edible fungi treasured by gourmet chefs worldwide. Some species are saprotrophic and others are able to form facultative mycorrhizal-like associations with plant roots without establishing true ectomycorrhizal symbioses. In general, it is considered that the formation of asexual spores, or mitospores, is an important step in the life cycle of morels. However, ultrastructure characterization and physiological attributes of morel mitospores have received little attention. In this contribution, the mitospores of *M. sextelata* were successfully induced under laboratory conditions and their ultrastructure, occurrence, germination, physiological characteristics and mating type gene structure were studied. Mitospore production was closely related to aeration, nutrition and humidity conditions. The average germination rate of mitospores on different media and under various induction stimuli was very low, with an average of 1/100,000. Based on the ultrastructure characterization, low germination rate, growth rate decline, rapid aging and mating genotyping, it was concluded that the mitospores of *M. sextelata* had lost their conventional function as conidia and might act more as mate sperm-like (gamete) structures. Thus, this study contributed to a deeper understanding of the life cycle of the economically and ecologically important morel fungal group.

## 1. Introduction

Morels, which belong to the genus *Morchella* in the Ascomycetes, Pezizales and Morchellaceae taxonomic categories, are culinary delicacies that are widely prized worldwide [1,2]. They have a high economic value because of their unique taste and rich nutritional value [2]. Several bioactive compounds that improve immunity, reduce blood pressure, and provide anticancer and antitumor properties were also identified in morel species [3,4,5]. Morel species also have great ecological relevance as saprotrophs or ectomycorrhizas. Dahlstrom et al. [6] conclusively demonstrated that some morel species are able to establish true ectomycorrhizas with mantle and Hartig net formations with *Pinus*, *Pseudotsuga* and *Larix*. Currently, it is considered that there are morel species that form mycorrhizal-like structures without establishing true ectomycorrhizal symbiosis and others are true soil-saprotrophic species, as demonstrated by their commercial cultivation in bare soil lacking symbiotic trees [2,7]. The domestication and cultivation of *Morchella* have a history of several hundred years; however, it was not until recently that their indoor and field cultivation were achieved [2,7,8]. However, due to the weak research on their biological basis, cultivation has suffered serious instability [7,9]. For example, indoor cultivation in the United States has stagnated after a period of operation, and field cultivation in China has recently also suffered 70% non-profit situations in some areas [9,10,11,12]. As a consequence, there has recently been a comprehensive promotion of field cultivation based on deep knowledge development of basic biological foundations, such as the identification and characteristics of cultivable varieties [12,13,14,15], the identification of mating type genes [9,10], the characteristics of heterokaryons [16,17,18,19,20,21], nutritional metabolism [22,23] and “omics” technology [19,24,25,26,27,28,29]. However, key questions related to the life cycle of *Morchella* remain far from completely understood [9,10].

*Morchella* can produce two types of spores in its life cycle: meiotic ascospores and mitotic mitospores. In general, the reproduction and transmission of fungi can be carried out via asexual (or mitospores) and/or sexual spores (or meiotic spores), which can be haploid, diploid or polyploid, with each one having their own characteristics in terms of morphology, size and ornamentation color. Fungal meiospores can be contained within the sac or on the basidium [30]. Generally, spores have a relatively static metabolism, and therefore, through quantitative advantages, including thickened cell walls, pigments and lipid storing, they can resist adverse living environments. Some types of mitospores (such as chlamydospores) are particularly highly resistant. Therefore, they function in some ascomycetes as dispersal units. Most of them are able to break their dormancy and germinate under appropriate growth conditions, as shown by the culturable fungal spores in marine sediments and Arctic ice after thousands of years [30,31]. Although spore production is regarded as the continuation of microbial life, the possibility of survival can be realized only when spores are able to restart their vegetative growth (germination) under appropriate conditions. Conidia, zoospores, arthrospores, rust spores and microconidia belong to the category of fungal mitospores, which is used as a descriptive morphological term, but do not include specific functional attributes [32]. It is generally believed that microspores play the role of immobile sperm, but in-depth research demonstrated than in many ascomycete fungi, including *Magnaporthe oryzae, Neurospora crassa* and *Hymenoscyphus fraxineus*, also have the transmission function of germinating into vegetative hyphae or infecting the host, which makes the functional definition of mitospore very complex [33,34,35].

In the case of *Morchella*, the haploid characteristics and genetic variation of the multinucleate homokaryon nature of ascospores were preliminarily studied [16,17,18,36]. Additionally, ascospore characterization is an important microscopic feature in the taxonomy of the genus [37,38,39]. However, in general, the study of *Morchella* mitospores has received little attention. The early research recorded on the mitospores of *Morchella* mostly came from observations during domestication and cultivation experiments and field ecological observations. Molliard first recorded the occurrence of mitospores during the morel domestication process conducted in 1904 [40]. In addition, Ower detailed the occurrence time of mitospores of *M. rufobrunnea* and briefly recorded the phenomenon that they were difficult to germinate [1,8]. Masaphy et al. (2010) carefully observed and recorded the morphology of mitospores during the cultivation of *M. rufobrunnea* [41]. Recently, Carris et al. (2015) described the natural conditions of *Morchella* mitospore production [42]. However, the deepest understanding of *Morchella* mitospore came from their field cultivation in China [2,43]. However, so far, there are only very few reports on the asexual sporogenesis and germination of *Morchella* under laboratory conditions. Baran and Boroń (2017) pointed out that some cultures of *M. deliciosa* form a small number of mitospores after 6 weeks of culture, while *M. esculenta* did not form them under the same conditions [44]. Yuan et al. (2019) induced the occurrence of mitospores of *M. sextelata* under indoor natural conditions (5 °C to 17 °C, day and night natural light conditions) and studied the morphological characteristics and germination characteristics [45]. Despite this, seminal knowledge regarding the occurrence conditions, germination characteristics, genotypes, physiological characteristics of mitospore germinated strains and key issues of the mitospore functions are in their infancy.

Therefore, this study was designed and conducted to contribute to filling in some gaps in the knowledge related to the structure and physiology of morel mitospores in order to contribute to a deeper understanding of the morel life cycle. In particular, the following questions were assessed: (i) Which factors are involved in the promotion of mitospore formation? (ii) What are the germination and growth rates of mitospores compared with their parental cultures? (iii) What were the ultrastructure and mating genotyping structure of the studied morel population?

## 2. Materials and Methods

### 2.1. Fungal Material

A commercial heterokaryon strain identified as *M. sextelata* No. 13 was used as the main experimental fungal material for all experiments. Additionally, heterokaryon strains of *M. importuna*, *M. eximia*, *M. galiliaea* and *M. crassipes* were used to evaluate their mitosporogenesis. All strains were identified using nrITS, LSU, *tef*1α, *rpb*1 and *rpb*2 multigenic phylogenetic analysis. The GenBank accession codes for the molecular markers nrITS, LSU, *tef*1α, *rpb*1 and *rpb*2 are as follows: *M sextelata* (OQ268248, OQ255581, OQ274136, OQ274141, OQ274146), *M. importuna* (OQ268249, OQ255582, OQ274137, OQ274142, OQ274147), *M. eximia* (OQ268250, OQ255583, OQ274138, OQ274143, OQ274148), *M. galiliaea* (OQ268251, OQ255584, OQ274139, OQ274144, OQ274149) and *M. crassipes* (OQ268252, OQ255585, OQ274140, OQ274145, OQ274150), respectively.

### 2.2. Induction Conditions of Mitospore Formation

The induction containers were 9 cm diameter plates with a space height of 1 cm or 1.5 cm, as well as 350 mL tissue culture bottles (with a diameter of 75 mm and a height of 108 mm). The basic culture medium used was 0.8–1.2% water agar, which was poured into the induction containers (either 15 mL into the 9 cm plates or 40 mL into the 350 mL tissue culture bottles). After the water agar solidified, sterile sawdust, straw, mushroom cultivation residue (originated by cultivation of *Pleurotus eryngii*), corncob, wheat grain or pine needles (3–6 cm long, 20–40 needles in each container) were added. The activated spawn was inoculated with 0.8–1.0 cm^2^ plugs from morel cultures and incubated at 18 °C. After that, regular observation and recording of the mycelial germination and the formation state of mitospore mats were recorded.

### 2.3. Mitospore Germination

Under sterile conditions, mitospore mats were picked up and placed in 2–3 mL containers with sterile water and shaken for 20–30 s. After that, the mycelial fragments were filtered out with an 8 μm aperture sterile microporous filter in order to obtain pure sterile mitospores. The mitospore concentration was evaluated in a blood cell counting plate by using a compound microscope (Leica M205, Wetzlar, Germany). The germination induction of mitospores was carried out using a variety of media and stimulation conditions.

The used media were CYM, PDA, MYG, SYM and CHM, with the culture media formulation as follows: CYM solid medium (L^−1^)—yeast extract 2 g, peptone 2 g, K_2_HPO_4_ 1 g, MgSO_4_ 0.5 g, KH_2_PO_4_ 0.46 g, glucose 20 g and agar 20 g; PDA solid medium (L^−1^)—boiled juice filtrate from 200 g of potatoes, glucose 20 g and agar 20 g; MYG solid medium (L^−1^)—malt extract 20 g, yeast extracts 1 g, peptone 1 g, glucose 20 g and agar 20 g; SYM solid medium (L^−1^)—KNO_2_ 10 g, yeast extract 2.5 g, glucose 20 g and agar 20 g; and CHM solid medium (L^−1^)—NaNO_2_ 3 g, K_2_HPO_4_ 1 g, MgSO_4_ 0.5 g, KCl 0.5 g, FeSO_4_ 0.01 g, glucose 20 g and agar 20 g. The stimulants included the addition of 1% of the following extracts: (i) Soil extracts obtained at two different times from *M. sextelata* cultivation in a field. In order to obtain these extracts, soil was sampled at a 5 cm depth either 20 days after sowing when the mitospores were very abundant or 40 days after sowing when the mitospores began to subside before ascomata primordium occurrence. The soil sampled in these ways was mixed with distilled water in a 1:1 ratio, soaked at room temperature for 1 h to obtain the soil extract and then filtered under axenic conditions with a 0.22 μm aperture filter to obtain a sterile extract. CYM basic medium was added with the 1% soil extract to induce the germination of mitospores. (ii) Liquid culture medium of *M. sextelata* cultivated in CYM for 7 days, which was previously passed through a 0.45 μm diameter filter. (iii) Sclerotia and intracellular mycelium extract produced as follows: First, sterile cellophane was placed on plates containing CYM solid medium; then, the plates were inoculated with activated *M. sextelata* spawn and cultured at 18 °C for 2 weeks to obtain mycelium containing sclerotium cells. After this, the sclerotium and mycelium cells were ground with 2× sterile distilled water, then filtered with a 0.22 μm aperture filter to obtain a sterile extract. (iv) Sclerotia and mycelial cell fragments were produced as follows: 3 g mycelial cell fragments generated as explained for the previous extract were placed in 30 mL of molten CYM solid medium at 45–50 °C, mixed evenly, and poured into 9 cm sterile plates, cooled, coated with active mitospores and cultured at 18 °C.

### 2.4. Microscopic Analysis

The culture bottles or dishes where the mitospores were cultivated were regularly observed under a stereomicroscope (Leica S8, Wetzlar, Germany) and photographed. Additionally, the mitospore mats were picked up with a fine needle and put on a 15% glycerol aqueous solution, photographed with a compound microscope (Leica M205, Wetzlar, Germany), and the spores, spore stalks and hyphae were measured by using the microscope software. Fifty structures were measured for each case. Morel mitospores were freeze-dried and subjected to scanning electron microscopy (Zeiss, Sigma 300, Oberkochen, Germany) in conjunction with cryo-preparation (PP3010T, Quorum, Lewes, UK). Briefly, the mitospore mats were adhered to the conductive tape and then frozen with liquid nitrogen, transferred into a cryo-preparation chamber and then sublimed at −90 °C for 10 min. Following the gold sputter-coating for 90 s at 5 mA, the samples were observed at 7 kV. Furthermore, a transmission electron microscopy analysis of the mitospore ultrastructure was conducted. In order to do this, mitospore mats were washed with PBS 2–3 times for 10 min each time, then fixed with 1% osmium acid for 2–4 h and washed with phosphate-buffered saline (PBS; pH 7.4). Subsequent to gradient dehydration with 30%, 60%, 75%, 85%, 95% and 100% alcohol for 15 min each time, the residual alcohol in the dehydrated sample was replaced with 100% acetone twice for 15 min each time. Then, the samples were embedded with SPI-Pon 821 (EMS, West Chester, USA). The samples were repaired, positioned and sliced ultrathin (Leica U7, Wetzlar, Germany) with a section thicknesses of 70 nm. The sections were double-stained with uranyl acetate and lead nitrate and then observed and analyzed using a transmission electron microscope (Hitachi HT-7700, Tokyo, Japan). Finally, the observation of mitospore nuclear behavior was carried out with the help of laser confocal microscopy (Zeiss LSM900, Oberkochen, Germany). Briefly, mitospore mats were first taken and placed in 1 μg/mL DAPI (4′,6-diamidino-2-phenylindole, Sigam) nucleic acid dye in the dark for 10 min of dyeing, and then carefully washed with PBS. After that, observation and photography were conducted after sectioning with 15% glycerol, and the number of nuclei in the mitospores was counted; the measurement was conducted with 200 mitospores.

### 2.5. Determination of Growth Rate and Longevity of Mitospore-Germinated Strains

Mitospore-germinated strains were initially incubated in CYM solid plates at 18 °C in order to evaluate their growth rate expressed in mm per h. Germinated strains obtained from pure sterile mitospores as described above were considered the parent cultures. The life span of these cultures was determined through subculture transfers (Appendix A). Briefly, 13 cm square Petri dishes were used and the strains were inoculated in one corner of these plates containing CYM solid medium, cultured at 18 °C and a record at the growing front of the colonies was registered every 3–4 days in order to document the growth speed of the colonies. When the colonies reached the opposite corner to the point where they were inoculated, an explant 0.4 × 0.8 cm was cut from the mycelial growing front and this explant was considered “the first generation”. Then this explant was located following the same method in a new plate, and the process continued until the colony growth stopped. The growth rate of each generation was then calculated, and the total culture duration and total growth distance were cumulatively calculated to evaluate the life span of each evaluated strain.

### 2.6. Mating Genes Detection of the Mitospore-Germinating Strains

Those strains where mitospore germination was detected were propagated in CYM liquid medium at 18 °C for 7 days. Then, the mycelium was sampled, washed twice with sterile water, dried with clean filter paper, and directly used for DNA extraction or stored at −80 °C for later analysis. For DNA extraction, the Plant Genome Extraction Kit (Tsingke TSP101-200, Beijing, China) was used following the instructions of the manufacturer. A nanodrop spectrophotometer was used to measure the quality and concentration of DNA, diluted to 50 ng/uL and stored at 4 °C for later analysis. The Internal Transcriber Spacer (ITS) region was amplified with the primers ITS1/ITS4 [46]. The mating primer design was based on the genome sequence data of *M. sextelata* strain No. 13. Mat1-1AF/R (GACTGGAATTCTATGACCCCA/GAACTCCTGGAATGTCTGTGA) was used to amplify the Mat1-1-1 gene fragment of *M. sextelata* with the product size 574 bp and an annealing temperature of 58 °C, and Mat1-2BF/R (CGGGGATTGGGAACAACGAT/GCGGCGAACAACATCTTCAG) was used to amplify the Mat1-2-1 gene fragment of *M. sextelata* with the product size 327 bp and an annealing temperature of 60 °C. The PCR protocol was as follows: 1 × T8 High-Fidelity Master Mix (Tsingke, Beijing, China), 10 ng forward and reverse primers, and 50 ng DNA template. The PCR reaction conditions were as follows: pre-denaturation at 95 °C for 2 min, followed by denaturation at 95 °C for 5 s, specific annealing temperature for 10 s, extension at 72 °C for 10 s, 34 cycles, and finally, extension at 72 °C for 5 min. The PCR products were verified using 1.2% agarose gel electrophoresis.

## 3. Results 

### 3.1. Induction of Mitospore Formation

The results showed that *M. sextelata* could hardly metabolize pure sawdust, straw or mushroom production residual medium and, as a consequence, only a very lax mycelium was observed without mitospore formation (Appendix A). *M. sextelata* mycelium metabolized corncob and wheat grain better, and a certain amount of sclerotia were observed on plates containing wheat grain (Appendix A). Despite this, in these two treatments, no mitospore formation was recorded. In contrast, *M. sextelata* was also able to moderately metabolize pine needles. Additionally, in this treatment, a small amount of mitospores was occasionally recorded when pine needles were added as the inducing factor (Appendix A).

When exploring the relationship between aeration and the formation of mitospores, we first chose Petri dishes, allowed either a 1 cm or 1.5 cm space height, and cultured them at 18 °C. The results showed that after 30 days of incubation, more mitospores were observed in the Petri dishes with a space height of 1.5 cm on both the plate walls and over the pine needles (Figure 1b,c). However, mitospores were very scarce or more frequently not formed in those plates with a space height of 1 cm (Figure 1a). Additionally, when an airtight sealing film was used to seal the plates, very scarce mitospores formed, regardless of the space height of 1.0 cm or 1.5 cm. In the case of the cultivation conducted in 350 mL bottles with 40 mL of 0.8% water agar (pH 7.5–8.0), to which sterile pine needles were added and cultured at 18 °C, a large number of mitospores were produced 20–25 days after incubation (Figure 1c,d). These results were more conspicuously recorded for the *M. sextelata*, *M. importuna* and *M. eximia* strains. In contrast, *M. crassipes* and *M. gallica* did not induce sporogenesis in our experimental conditions despite many attempts.

### 3.2. Morphological Characteristics of Mitospores

Under laboratory conditions, mitospore production began to be evident 20 to 25 days after mycelial inoculation (Figure 1b–d). Optical and scanning electron microscopy showed that mitospores mostly formed in clusters at the tip of phialides supported by metulae. These phialides were usually formed in a complex system of branches (Figure 2a–c). Interestingly, in addition to the mitospore formed on phialides, a small number of asexual spores were recorded as produced directly on the hyphal surface. The mitospores were mostly spherical or subspherical with a diameter of 3.4–4.5 μm, whitish and translucid (Figure 2d–g). The diameters of metulae were 6.5–10 μm, and they were whitish and translucid. Ultrastructural microscopy allowed for the observation of subcellular organelle structures, including nuclei, mitochondria, endoplasmic reticula, vacuoles and lipid droplets. The cell wall was homogeneous with an average thickness of 100.75 ± 13.89 nm (minimum = 80 nm, maximum = 140 nm) (Figure 2h,i).

Nuclear staining showed that most mitospores were mononucleated (90.21%); the nuclei were usually rounded and mostly located in the middle of the mitospore (Figure 3a,b). However, bi- and multinucleated mitospores were also recorded in a minor proportion (9.79%) (Figure 3c,d).

### 3.3. Germination Characteristics of Mitospores

The mitospore germination of *M. sextelata* was only recorded when the mitospore concentration was equal to or higher than 10^4^ per dish. Different culture media had slightly different effects on the germination induction, but the germination rate was always very low, with the average being 1.22 × 10^−5^ (ranging from 5.79 × 10^−5^ to 2.64 × 10^−6^). The SYM culture medium presented the lowest germination, followed by CYM and MYM, while the PDA and MYG culture media had the highest germination rates (Table 1). It should be noted that the extracellular secretion of CYM liquid culture mycelium, sclerotium and mycelium cellular inclusions and fragments, and soil extract at different stages of field cultivation of *M. sextelata* did not promote the germination of mitospores. At the initial stage of the mitospore germination, the hyphae were thin, and each germination hypha was clearly visible; then, an abundant mycelial growth was observed and a slightly dense colony was formed until the diameter of the colony was about 1 cm. The germinated colonies were transferred to the new PDA plates. Except for a small number of colonies that died quickly after being transferred (in these cases an increase in dark pigments was observed, along with a growth rate decrease, followed by death), the morphology of colonies was similar to the conventional *M. sextelata* colonies previously reported. In 63 experimental treatments with 8 replicates, only 108 mitospores germinated, and from them, only 94 survived (Table 1). The earliest germination time was 6 days after culture, and the average value was 14.24 days after culture.

### 3.4. Growth Rate and Longevity of Mitospore-Germinated Strains of M. sextelata

There were 94 out of 108 germinated mitospore strains that survived after the transfer, all of which were verified using ITS sequencing to confirm that they were consistent with their parent *M. sextelata* strain. When these germinated strains were cultured, it was found that their growth rate, pigment and colony morphology presented some variation. Then, the growth rate of all germinated strains and the life span of 38 randomly selected germinated strains were determined in detail. In terms of the mycelial growth rate, most strains (82.11%) presented a lower mean growth rate (0.53 ± 0.03 mm/h) than that of their parental strains, which was 0.62 ± 0.05 mm/h. Only 17 germinated strains (17.9%) had a higher mean growth rate than that of their parental strains. Some strains showed very low growth rates, such as AH03, AF14 and AH05 (Figure 4).

The longevity test results are shown in Figure 5. The longevity (culture duration and total growth distance) of most mitospore-germinated strains was significantly lower than that of the parental strain. The parental strain was a commercial strain with high vitality, which reached accumulated subculture values of 4779.5 h and a total accumulated length of 271.2 cm. Among the 38 mitospore germinated strains, only 6 of them (AF08, AF11, AF13, AI20, AI21, AI22 and AI24) were cultured for longer than the parental strain in terms of accumulated time when adding the subcultivation process described above (Figure 5 and Appendix A). However, the total accumulated growth distance was lower in all evaluated strains than that of their parental strain (Figure 5). For example, AI24 presented the longest accumulated culture time when adding the total subculturing, reaching 5558 h, but its total growth distance was only 149.07 cm, which represented only 54.97% of the parental strain’s growth distance. Among the 38 strains tested, 73.68% (28 out of 38 strains) grew for less than half of the total accumulated time than that of their parental strain. In terms of the total accumulated growth distance through subculturing, 89.47% had a total growth distance lower than half of its parental strain (Figure 5). The physiological degradation of *M. sextelata* mitospore-germinated strains might indicate their weak viability. Combined with the extremely low germination rate, it was speculated that they might no longer have the traditional transmission and reproduction function like conidia.

### 3.5. Mating Type Genotyping of Mitospore-Germinated Strains

It was unknown whether the multiple nuclei in the mitospores originated via direct migration from the sporulation structures or mitosis after the formation of mononucleate mitospores. Thus, in order to answer this question, mating type gene detection on all 94 germinated strains was conducted. The results showed that two sets of mating type primers could successfully amplify clear target bands, and all strains could amplify at least one target band, including 64 strains containing the Mat1-1 mating type, accounting for 68.09%, and 42 strains containing the Mat1-2 mating type, accounting for 44.68%. Meanwhile, 12 strains contained both the Mat1-1 and Mat1-2 mating type genotypes, accounting for 12.77% of the population (Figure 6). Strains containing only one single mating type (Mat1-1 or Mat1-2) genotype accounted for 87.23%, which was approximately the same as the 90.21% mononuclear proportion observed using a confocal microscope (Figure 3). Therefore, it was demonstrated that 12.77% of the total evaluated population, i.e., those strains that contained both mating genotypes and mitospore multinucleation occurred via direct migration from the sporulation structures rather than mitosis after the mononuclear mitospore formation.

## 4. Discussion

### 4.1. Influence of Mitospore Formation by Nutrition, Aeration and Humidity

The sporulation of fungi is a complex process that involves not only the influence of environmental conditions but also the control of internal biological rhythms [47,48]. Studies on the molecular mechanism of sporulation in *Aspergillus nidulans* and *Neurospora crassa* revealed that a large number of genes affect the different steps of spore formation [49,50]. Meanwhile, many interdependent events must interact in space and time to induce spore formation [31,50]. In short, spore formation usually occurs when the growth state is inhibited, for example, by starvation or nutrient deficiency. Therefore, low nutritional artificial media are usually used to induce the production of mitospores [48].

Generally, in the field cultivation of morels, the mycelium grown on the ground surface can begin to change into mitospores 7–10 days after sowing. Over time, about 15–20 days after sowing, the mycelium on the ground surface will have abundant mitospore formation. However, the mycelium network in the soil or the surface mycelium under the condition of local hypoxia caused by the use of mulching film did not change its morphology and continued to maintain its mycelial state [2]. Therefore, it was hypothesized that the mitospore formation might be related to aeration. Under laboratory conditions, mitospores could hardly be formed using tubes or plates in the conventional culture medium, even when aeration was sufficient (it could only be formed occasionally after long-term cultivation). Therefore, it was also hypothesized that nutrient enrichment could inhibit mitospore formation. In order to know the potential mitospore triggers, low-concentration water agar (0.8–1.2%) as a basic medium in plates was used, to which different sources were individually added, namely, sterile sawdust, straw, mushroom production residual medium, corncob, wheat grain and pine needles.

In the field cultivation of *Morchella*, 7–10 days after sowing, the mycelium growing on the soil surface began to metamorphose and form mitospores that showed a white, powdery appearance, which was called “fungal frost” (Appendix A) due to its similarity with the white frost on a winter morning [2,43]. In our research, at the initial stage, only vegetative hyphae were observed, then metamorphosed to form small clusters and white powdery mats started, and gradually larger mitospore aggregates were recorded. The formation of mitospore mats was first related to other fungal species. At present, *M. importuna*, *M. sextelata*, *M.exima* and *Mel-21*, which are commonly cultivated species, frequently form this kind of mitospore mats, while most yellow species, such as *Mes-6*, *Mes-19* and *M. spongia*, did not present them, even in the same field environments. In this study, mitospore induction experiments of several species were carried out at the same time, and the results were consistent with the cultivation experiments. The induction of mitospore formation in *M. importuna*, *M. sextelata* and *M. exima* was easily induced, while in the case of the yellow morel varieties of *M. crassipes* and *M. galiliaea*, this phenomenon could not be induced. The formation of mitospore mats was also influenced by aeration. In order to maintain the soil humidity and temperature, plastic film mulching was usually used after sowing. In the areas with poor aeration, due to tight film coverage, mitospore mat formation was significantly reduced [2], which should also be a reason why it was difficult to induce mitospore formation under laboratory conditions. In this study, we successfully induced mitospores with a large space and breathable membranes by ensuring sufficient aeration. A third factor that was found in our research was that rich nutrition inhibited the occurrence of mitospores. *Morchella* spp. strains are able to easily use a variety of sugars with low molecular weight, but it is more difficult for them to use more complex compounds, such as lignin and cellulose [23]. In low-fertility cultivation conditions, the growth of vegetative hyphae depended entirely on the nutrients contained in basic culture media or the original inoculation blocks, and the hyphae were weak and unable to form mitospores. In contrast, in the nutrient-rich wheat grain and corncob medium, the nutrient hyphae were abundant but the formation of mitospores was still inhibited (Appendix A). The same phenomenon was observed in traditional conventional culture media (CYM, PDA). The observation carried out in our research showed that *M. sextelata* mycelium metabolized corncob and wheat grain better than straw or mushroom production residual mycelium, which agreed with early research results showing that the starch in wheat grain and the available sugars in corncobs were easily metabolized by *Morchella* [2,23,51]. Additionally, pine needle medium was found to be very effective in the sporulation of plant pathogenic fungi, including in the genera *Colletotrichum*, *Cryphonectria*, *Phomopsis*, *Fusarium*, *Phyllosticta* and *Mycoleptodiscus*. Out of the 42 strains evaluated belonging to these genera, 14 were found to form mitospores when grown in a pine needle medium [48]. Among fungi, extremely poor access to nutrients is an important condition that induces the production of mitospores, such as low-concentration PDA or just water agar [48]. In addition to the above reasons, the strain growth stage is also an important factor in the formation of mitospores; it is difficult to induce them from strains with low growth activity. Despite the fact that we did not conduct systematic evaluations, it was clearly observed that in those strains that were in a low growth stage, mitospore formation was less abundant compared with those having more active mycelial growth. It should be noted that air humidity was also an influencing factor for mitospore formation. When the air humidity was insufficient, the mycelium had poor growth and particularly poor aerial mycelium. Even if the nutrition and aeration were sufficient, mitospore production could not be induced. Regarding the wild natural environment, Healy et al. (2013) pointed out that the dry environment was not conducive to mitospore formation when investigating and analyzing the wild environment of naturally occurring mitospores in Pezizomycetes [52]. The formation of mitospores on hyphae surfaces observed in our research constituted the first report of this kind of growth in *Morchella* (Figure 2d–g). Previously, this phenomenon was only recently recorded in another Ascomycete, namely, *Tuber japonicum* [53]. The reported homogeneous cell wall, with a medium thickness of 100.75 ± 13.89 nm (Figure 2h,i), was much thinner than those reported for hyphae and sclerotia cell walls [15,54]. The mono- and multinuclear states of the strains obtained from the germinated mitospores observed in this study were consistent with those of the *M. importuna* and *M. sextelata* reported in earlier studies [43,45].

### 4.2. Mitospore Germination Conditions and Rapid Aging

Using monokaryotic strains obtained from single ascospores induced the production of mitospores of *M. sextelata* under laboratory conditions, Yuan et al. (2021) pointed out that these mitospores could germinate on PDA medium 1–2 days after inoculation originating germination tubes and then forming mycelium [45]. However, in this study, we found different behavior: (i) The germination time was slow. The average earliest germination time started 14.22 days after incubation, which was significantly later than Yuan’s report, where the germination started from a few hours to 2 or 3 days after inoculation. (ii) The germination rate was very low. The difference with Yuan’s experimental results may be related to interspecific differences between the strains. Molloard (1904) recorded that the germination of *M. esculenta* mitospores could not be achieved on a variety of media [40]. It was also simply mentioned that “the germination rate was less than 1% on various media” [8]. As far as we know, there has been no other report on the mitospore germination of *Morchella*. Fukumori et al. (2004) showed that there was no germination of the microconidia of *Botrytis cinerea* at a concentration of 10^4^ per dish on a variety of media [55]. A similar difficulty to germinate mitospores was reported in other species of Pezizomycetes [52]. In this study, the average germination rate of mitospores of *M. sextelata* was only 1/100,000. Similarly, Zhang et al. (2014) pointed out that 83% of germinated strains of *Magnaporthe oryzae* could not further form normal colonies compared with their parental strains [35]. The recorded germination time of mitospores in our research was much slower compared with that of sexual ascospores or tissue isolation, which usually occur between 2–3 days after inoculation [2,56,57].

When mitospores are used as an efficient reproductive strategy to spread fungal growth and resist adverse living environmental conditions, they need to germinate quickly in new suitable environments to promote the rapid propagation and reproduction of their genetic material. In order to carry out this successful strategy, their structure and physiological characteristics suffer fundamental changes, such as modifications in their shape, size and thickening of cell wall structure, as well as increases in their melanin cell wall and thorn production on the cell surface, which produce high germination rates [30,43,49,58]. The mitospores of *Morchella* were similar to some spherical conidia in morphology, but their other structural characteristics were very different from the described changes that occur in other taxa. For example, the thickness of the cell wall was very thin, homogeneous and lacking an outer layer [59,60]. Additionally, their abiotic germination conditions and rates were very specific and low, respectively. At the same time, the growth rate of most germinated strains and the longevity (total accumulated growth distance and total growth duration) were all significantly weaker than those of their parental strain. This led us to speculate that they lost the ability of traditional conidia, or at least that spreading and reproduction was not their main function.

### 4.3. The Mitospore of M. sextelata Should Be a Gamete

The morphological and physiological characteristics of the mitospores of *M. sextelata* were similar to those of the microconidia of most ascomycete fungi: small cell structures, thin homogeneous cell wall structure, no pigments in the outer layer of the cell wall and mononuclear nature [2,33,58,59,60,61,62]. Several previous studies showed that the sexual spores of *Morchella* might had multinucleate homokaryon states [9,10,17,18]. In physiological terms, most microconidia play the role of sperm gametes [33,55,63]. In this latter case, multinucleated mitospore formation might be explained due to (i) nucleus migration directly from the sporogenous structures at the beginning of mitospore formation or (ii) the occurrence of a second mitosis in the cells after mitospore formation. Bistis (1981) first analyzed the process of *N. crassa* microconidia attracting and then interweaving the trichogyne via in situ observation, and speculated that the microspores could secrete pheromones to induce the occurrence, movement and fusion of trichogynes [63]. The occurrence of ascocarps can be successfully induced by the contact between the microspore produced by *B. cinerea* and their sclerotia [55]. Theoretically, the microspores that play the function of the sperm should lose their potential for propagation and germination, while the microspores of some ascomycete fungi still have high germination efficiency in order to form vegetative hyphae, such as *Hymenoscyphus fraxineus* (~6.3%) [34], *Magnaporthe oryzae* (approximately 10%) [35] and *N. crassa* (1% to 95%) [33], which makes the functional boundary of these mitospores blurred [32].

Except for the recent observation of nuclear distribution and meiosis in the ascomycetes, there are no reports on cell structures on any possible involvement in the fertilization process of *Morchella* [18]. *Morchella* forms large ascomycete sporomes. Nuclear fusion and meiosis occurred in the hymenium of mature or nearly mature ascocarps, but when and where complementary mating nuclei (or cells) migrate (or fuse) into a cell was unknown. Different from the sclerotia of *Sclerotinia sclerotiorum* and *B. cinerea*, the sclerotia of *Morchella* spp. is just a mass of mycelium that is twisted and interwoven, without an obvious cortex. In terms of function, they were considered to play a dual role in storing nutrients and resisting adverse environments [1,54]. Jiang et al. (2021) pointed out that some sclerotia of *Morchella* only had one mating type gene [64], but in fact, both homokaryon and heterokaryon hyphae can form sclerotia [54,65]. This generates uncertainty regarding whether the sclerotia of *Morchella*, such as *B. cinerea* and *S. sclerotiorum*, can accept complementary mating gametes to complete the heteronucleation process. Shi et al. (2019) reported that there are transparent thin-walled hyphae at the base of the primordia of *M. sextelata* [66]. Whether these thin-walled hyphae occur before or after the primordia formation or whether they act as fertilizing trichogynes remains unclear. In this study, we found that different culture media and substrate additions, including fragments of mushroom-cultivated mycelia and sclerotium cells, had no effect on the germination of mitospores, which implied that the mitospores more likely functioned as gametophytes. If fertilization occurs, the most likely time–space stage might be between the sclerotia and the primordia formation. However, when and if this occurs needs further exploration.

## 5. Conclusions

Combined with the field production experience, this research successfully achieved the formation of mitospores of *M. sextelata* under laboratory conditions. The mitospore formation of *M. sextelata* was mainly related to poor nutrition, sufficient aeration and proper air humidity. Additionally, it was demonstrated that pine needles were able to significantly promote the morel mitospore formation. Furthermore, our research contributes to the understanding of the formation process of mitospores by using a clear field compound, as well as scanning and transmission electron microscopy. It was found for the first time that the morel mitospores were able to be directly produced over lateral hyphae, in addition to the more typical production on phialides supported by metulae. In terms of subcellular structure, it was observed that mitospore cells contained complete organelles, including nuclei, mitochondria, endoplasmic reticula, lipids and vacuoles. Ultrathin sections also showed that the cell wall of the mitospore was homogeneous and very thin with a thickness of about 100 nm. Fluorescence staining and confocal microscope observation of the mitospores showed that around 90% of the mitospores only contained one nucleus, and the rest had two or more nuclei. The germination rate of the mitospores was very low, being around 1/100,000 in various media and induction conditions. The mitospore germination strains showed quick aging compared with their parental strains. The mating type analysis showed that around 87% of the mitospore germination strains only contained one of the two mating type genotypes (Mat1-1 or Mat1-2), but the remainder contained both, which indicated that the multicellular nuclear state in the mitospore directly migrated from the sporogenous stem rather than the new mitosis after the formation of mononuclear mitospores. Based on the previous results, we concluded that the morel mitospores had lost their traditional function as conidia and were more likely to act as spermatia, just like microspores.

## Figures and Tables

**Figure 1 microorganisms-11-00345-f001:**
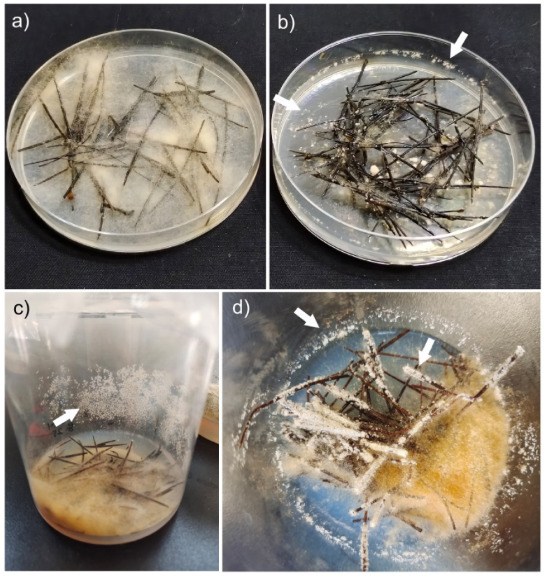
Induction conditions of mitotic sporogenesis. (**a**) Conventional cultivation in a 1 cm-high plate without the presence of mitospores. (**b**,**c**) Abundant mitospore formation in a 1.5 cm high cultivation plate (**b**) and a 10 cm height culture bottle (**c**). (**d**) Close-up picture showing abundant mitospore mat formation (arrowed) on both the container walls or over the pine needle surface.

**Figure 2 microorganisms-11-00345-f002:**
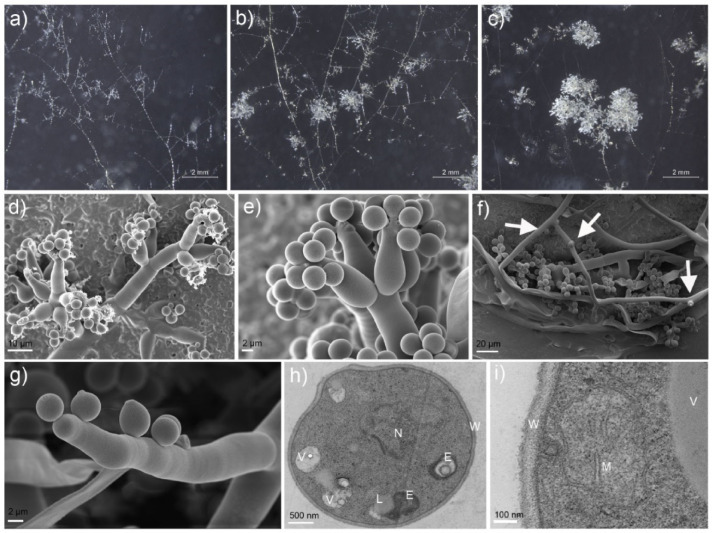
Formation and microscopic characteristics of *Morchella sextelata* mitospores. (**a**) Early state of mitotic spore development. (**b**) Mature abundant mitospore mats. (**c**) Close-up of mature mitospore mats appearance. (**d**–**g**) SEM images of mitospores: (**d**,**e**) mitospores growing from phialides supported by metulae, and (**f**,**g**) mitospores directly produced on hyphae surface (arrows). (**h**,**i**) TEM observation of mitospore inner structures: E—endoplasmic reticulum, L—lipid droplet, M—mitochondrion, N—nucleus, V—vacuole and W—cell wall.

**Figure 3 microorganisms-11-00345-f003:**
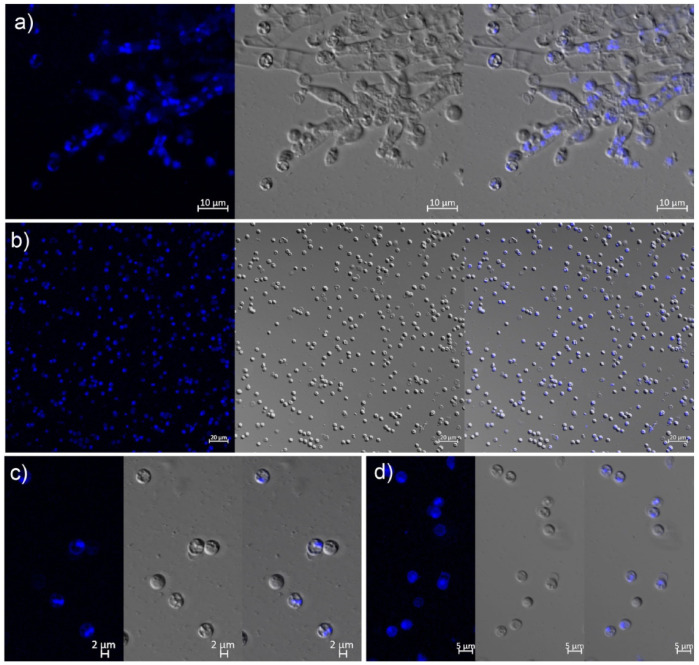
Morphological appearance of morel mitospores under laser confocal microscopy. (**a**) Mitospores growing from phialides supported by metulae. (**b**) Mitospores masses showing abundant mononucleate states. (**c**,**d**) Occurrence of bi- (**c**) and multinucleate (**d**) mitospores.

**Figure 4 microorganisms-11-00345-f004:**
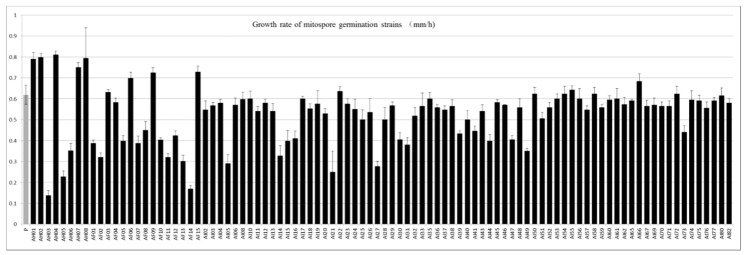
Average growth rate of the parental strains and 94 mitospore-germinated strains. P represents the mean growth rate of the parental strains, while the AH, AF and AI series represent the mean growth rates of mitospore germination strains in different batches.

**Figure 5 microorganisms-11-00345-f005:**
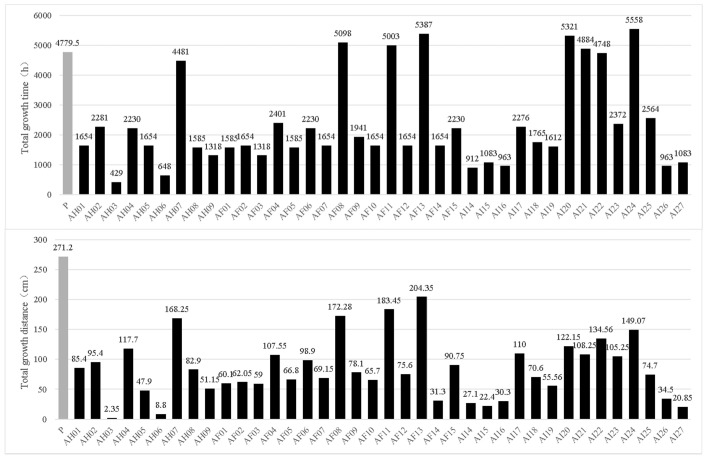
Longevities of the 38 mitospore-germinated strains of *M. sextetala* compared with those of their parental strains (P), shown as the total accumulated growth time (**above**) and total accumulated growth distance (**below**) after subculturing.

**Figure 6 microorganisms-11-00345-f006:**
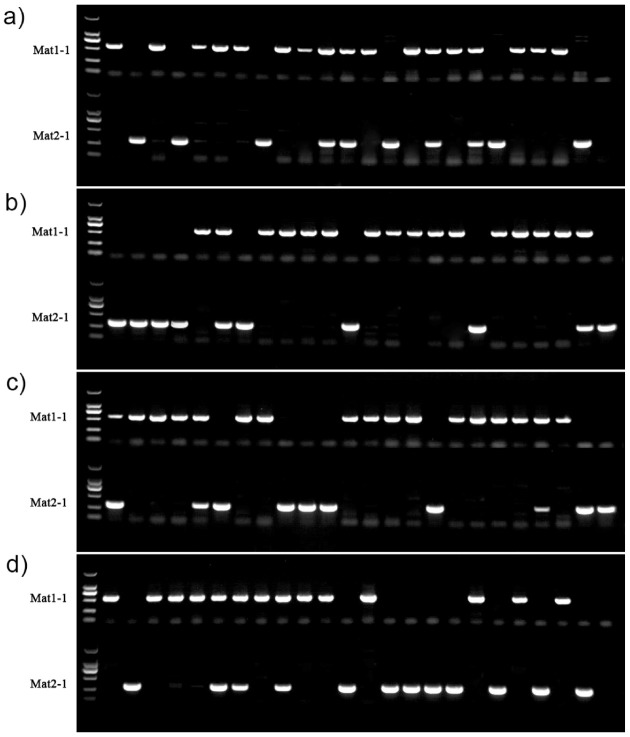
Genotyping results of the mating type of 94 germinated strains of *M. sextelata*. The leftmost lane of all pictures (**a**–**d**) was the DL2000 molecule marker. (**a**) Lanes 2 to 9 represent the 8 strains of the AH series and lanes 10 to 23 represent the 15 strains of the AF series. (**b**,**c**) Lanes 2 to 24 and lanes 2 to 23 (**d**) represent the 71 strains of the AI series. Mat1-1 and Mat2-1 represent two different mating genotypes.

**Table 1 microorganisms-11-00345-t001:** Germination condition and state of mitospores of *Morchella sextelata*.

Mitospore Properties	Culture Medium	Initial Concentration	Number That Germinated	Germination Rate	The Earliest Germination Time	The Latest Germination Time
Mitospores were kept at 4 °C for 1 month	SYM	2.65 × 10^5^				
	PDA	2.65 × 10^5^	5	1.88679 × 10^−5^	11 days	39 days
	CYM	2.65 × 10^5^				
	MYM	2.65 × 10^5^	2	7.54717 × 10^−6^	12 days	20 days
	CHM	2.65 × 10^5^	2	7.54717 × 10^−6^	14 days	59 days
Mitospores were kept at 15 °C for 2 months	SYM	1.34 × 10^5^				
	PDA	1.34 × 10^5^	1	7.46269 × 10^−6^	9 days	9 days
	CYM	1.34 × 10^5^				
	MYM	1.34 × 10^5^	3	2.23881 × 10^−5^	13 days	30 days
	CHM	1.34 × 10^5^	1	7.46269 × 10^−6^	36 days	
Mitospores were kept at 4 °C for 2 months	CYM	4.92 × 10^6^	13	2.64228 × 10^−6^	12 days	24 days
	CYM	4.92 × 10^5^				
	CYM	4.92 × 10^4^				
	PDA	4.92 × 10^6^	16	3.25203 × 10^−6^	9 days	25 days
	PDA	4.92 × 10^5^	4	8.13008 × 10^−6^	12 days	33 days
	PDA	4.92 × 10^4^				
	MYG	4.92 × 10^6^	21	4.26829 × 10^−6^	10 days	30 days
	MYG	4.92 × 10^5^	2	4.06504 × 10^−6^	20 days	30 days
	MYG	4.92 × 10^4^				
Mitospores were kept at 15 °C for 3 months	CYM	7.58 × 10^5^	2	2.63852 × 10^−6^	7 days	12 days
	CYM	7.58 × 10^4^				
	PDA	7.58 × 10^5^	7	9.23483 × 10^−6^	7 days	21 days
	PDA	7.58 × 10^4^	2	2.63852 × 10^−5^	17 days	23 days
	MYG	7.58 × 10^5^	15	1.97889 × 10^−5^	6 days	59 days
	MYG	7.58 × 10^4^				
Mitospores were kept at 4 °C for 7 months	SYM	2.46 × 10^5^	2	8.13008 × 10^−6^	18 days	21 days
	PDA	2.46 × 10^5^	2	8.13008 × 10^−6^	17 days	32 days
	CYM	2.46 × 10^5^	2	8.13008 × 10^−6^	18 days	
	MYM	2.46 × 10^5^	3	1.21951 × 10^−5^	15 days	19 days
	CHM	2.46 × 10^5^	0	0	Nonexistant	Nonexistant
Mitospores were kept at 4 °C for 7 months	SYM	3.45 × 10^4^	0	0	Nonexistant	Nonexistant
	PDA	3.45 × 10^4^	0	0	Nonexistant	Nonexistant
	CYM	3.45 × 10^4^	0	0	Nonexistant	Nonexistant
	MYM	3.45 × 10^4^	0	0	Nonexistant	Nonexistant
	CHM	3.45 × 10^4^	2	5.7971 × 10^−5^	6 days	13 days
Mitospores were kept at 4 °C for 9 months	CYM	9.17 × 10^4^	0	0	Nonexistant	Nonexistant
	PDA	9.17 × 10^4^	0	0	Nonexistant	Nonexistant
	MYG	9.17 × 10^4^	1	1.09051 × 10^−5^	30 days	30 days

## Data Availability

Accession numbers of DNA sequences are given in Materials and Methods section, Section 2.1.

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
