# Peer review of "Ultrastructure and Physiological Characterization of Morchella Mitospores and Their Relevance in the Understanding of the Morel Life Cycle"

_microorganisms, 2023, doi:10.3390/microorganisms11020345_

Round 1

Reviewer 1 Report

The authors reported new method in which mitospores are artificially produced by using pine needles, and revealed the part of life cycle of the economic and ecologically important mushroom, Morchella.

So, I think that this manuscript is valuable for the publication of Microorganisms. But I would like to recommend to minor revision to emphasize your new findings.  The revision points are listed as follow. Your consideration would be appreciated.

Line 299: droplets >>droplet

Line 299: mitochondoria >> mitochondorion

Line 300: vacuoles >> vacuole

Line 311: Please insert explanation regarding Fig.3 (d) in the figure legend.

Author Response

Microorganisms MDPI

Manuscript Number: microorganisms-2145717  

Ultrastructure and physiological characterization of Morchella mitospores and its relevance in the understanding of morel life cycle

We deeply appreciate the valuable corrections, suggestions and comments from the assigned editor and the two reviewers. In the updated version, now we have taken into account all the suggestions which in our humble opinion substantially enhance the manuscript. Taking into account the editor´s decision we have included now our specific answers in a table answering point by point. Additionally, we have highlighted with yellow text in the updated manuscript the additions or corrections we have made to answer the reviewers’ requests. Thanks a lot for your professionalism dealing with our manuscript.

Reviewer 2 Report

This manuscript is  interesting and gave an important contribution on a better comprehension of Morchella life cycle. It is also clearly written although it contains several imprecisions and some methodological details and results are lacking. In the results some sections are  part of the discussion.

The following parts of the manuscript need to be improved and some additional information should be supplied:

Line 19- It is not still not so clear which type of symbiosis the Morchella spp. establish with the hot plant. In my opinion they are not true mycorrhizal fungi.

Line 30 I don’t like  to much the tem microspores I prefer microconidia and spermatia.

Line 43. Dahlstrom et al. did not conclusively demonstrated that some morel species are able to establish true ectomycorrhizas. In the article, it is written that they form mycorrhizal like structures. I did some unpublished experiments in vitro conditions and they do not form  true mycorrhizas in my opinion.

Line 65. From my knowledge, fungal meiospores can be contained within a sac and on basidium

Line 67 The high resistance is a characteristic of most of the meiospores. Only few types of mitospores (like the chlamydospores) are highly resistant.  In many ascomycetes they function as dispersal units.

Line 75 I use the term megaspores for plant spores.

Line 77 I prefer to use microconidia rather than microspores.

Line 177 you have to give a reference of the article where the strains were identified by ITS-RPB1-117 RPB2-LUS-EF1α multigenic phylogenetic analysis. You have also the indicate the accession numbers of the previous published strains used in this work.

 In alternative (if your strains are new isolates) you have to report here the M &M and the results of these analyses including the new deposited accession numbers.

Line 170 “for sectioning” for the compound microscope? The sectioning procedures for TEM are reported in the lines 183-186.

Line 212 it is  a repetition” were propagated in CYM  liquid medium and cultivated in CYM liquid medium”

Lines 235-249 This is a part of the discussion

Line 248 What about the mycelia of the strains of the others species?

Lines 275-283 This is a part of the discussion

Lines 303-307 are part of the discussion

Line 322 in M & M (lines 196 and 212) you say that the germinated strains were cultivated in CYM

Line 410 Maybe you mean “The formation of mitospore mats was first related to - other  fungal species”?

Line 416-417 Most of the results are only on M.sextelata. No results about M. crassipes, and M. galilica ere reported. Add this information at line 268

Line 445 “the strain growth stage has also been  an important factor in the formation of mitospores” I can not find this result in  the result session

Line 457 “The mono- and multinuclear state observed in this paper” of the strains obtained from the germinated mitospores?

Moreover, no results were reported on M. importuna strains obtained from  germinated mitospores

Line 461 the article of Yuan et al. was published in 2021 not in 2019!

Yuan, B.-H.; Li, H.; Liu, L.; Du, X.-H. Successful induction and recognition of conidiation, conidial germination and chlamydospore formation in pure culture of Morchella. Fungal Biol 2021, 125, 285-293, doi:10.1016/j.funbio.2020.11.005

Yuan et al. 2021 used monocariotic strains obtained from single ascospores, the main strain used in your work was a heterocariotic strain. Moreover, they showed a conidia formation induction in cross-mating experiments. You may consider also these aspects

Lines 505 and 508 I prefer “microconidia”

Line 560 I prefer spermatia

Author Response

(The authors gave the same response as above.)

Round 2

Reviewer 2 Report

In my opinion the correct version of the manuscript could be accepted for publication after having correct this sentence

Fungal mitospores can be contained within the  sac, on the basidium, or directly produced from specialized sporogony on vegetative hypha, in isolated form, chains or clusters [30].

Fungal mitospores are not within a sac or on a basidium. That is a characteristic of fungal meiospores

Author Response

Microorganisms MDPI

Manuscript Number: microorganisms-2145717  

Ultrastructure and physiological characterization of Morchella mitospores and its relevance in the understanding of morel life cycle

We deeply appreciate the valuable correction from the reviewer 2. We have corrected the sentence requested by this reviewer. Thank you very much for your valuable help during the editorial process which enhanced the manuscript quality, originally submitted.

Question (Round 2) :The reviewer commented: In my opinion the correct version of the manuscript could be accepted for publication after having correct this sentence

Fungal mitospores can be contained within the  sac, on the basidium, or directly produced from specialized sporogony on vegetative hypha, in isolated form, chains or clusters [30].

Fungal mitospores are not within a sac or on a basidium. That is a characteristic of fungal meiospores

Answer (Round 2): We highly apprecciate the valuable correction. Then, we have corrected the sentence.

Following your advice we have now included the sentence: “Fungal meiospores can be contained within the sac or on the basidium”
